# Unveiling the Power of Argument Arrangement in Online Persuasive Discussions

**Nailia Mirzakhmedova**
Bauhaus-Universität Weimar

**Johannes Kiesel**
Bauhaus-Universität Weimar

**Khalid Al-Khatib**
University of Groningen

**Benno Stein**
Bauhaus-Universität Weimar

## Abstract

Previous research on argumentation in online discussions has largely focused on examining individual comments and neglected the interactive nature of discussions. In line with previous work, we represent individual comments as sequences of semantic argumentative unit types. However, because it is intuitively necessary for dialogical argumentation to address the opposing viewpoints, we extend this model by clustering type sequences into different argument arrangement patterns and representing discussions as sequences of these patterns. These sequences of patterns are a symbolic representation of argumentation strategies that capture the overall structure of discussions. Using this novel approach, we conduct an in-depth analysis of the strategies in 34,393 discussions from the online discussion forum Change My View and show that our discussion model is effective for persuasiveness prediction, outperforming LLM-based classifiers on the same data. Our results provide valuable insights into argumentation dynamics in online discussions and, through the presented prediction procedure, are of practical importance for writing assistance and persuasive text generation systems.

## 1 Introduction

Convincing others can be quite challenging, even when equipped with a comprehensive set of arguments. The questions then arise: what kind of arguments are the most convincing and which should be presented first? Should one begin with facts or personal experiences? The different answers to these and related questions are referred to as *argumentation strategies* (Al-Khatib et al., 2017). Several studies have empirically examined the arrangement of argumentative discourse unit (ADU) types, such as facts or testimonies, in monological texts (e.g., Al-Khatib et al. (2017)), or in individual comments within dialogues (e.g., Hidey et al. (2017); Morio et al. (2019)). These studies have shown that the

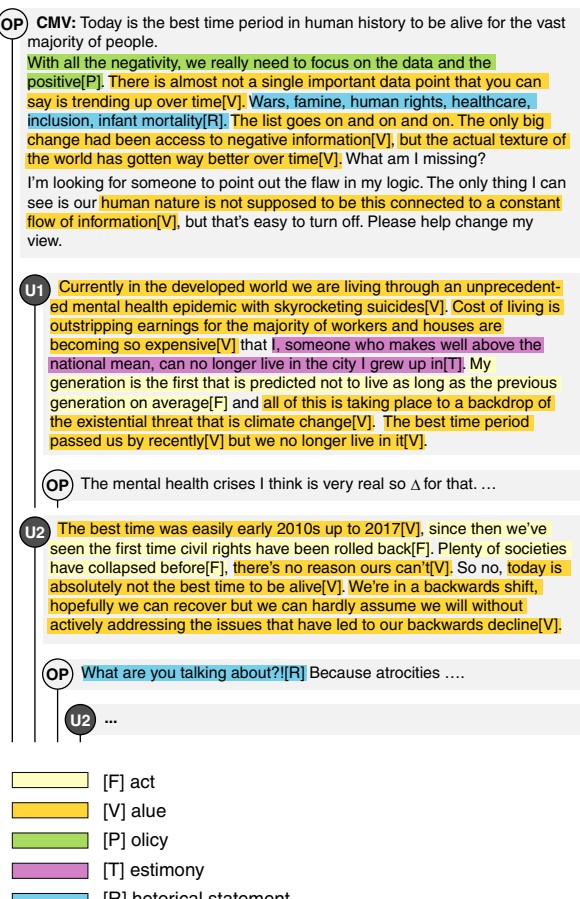

Figure 1: Illustration of one Change My View thread with two discussion branches (with commenters U1 and U2, respectively) and detected ADU types. The original poster's (OP) comment with Δ in the first discussion branch marks that branch as persuasive.

arrangement of ADU types in a text can serve as a model of the argumentation that underlies the text, for example for predicting its persuasiveness.

In dialogues, such as the two discussion branches shown in Figure 1, it seems intuitive that there is no single best argumentation strategy for either side. Instead, the strategy needs to be dynamically adapted in response to the ongoing conversation. For instance, if the opponent concludes their ar-

gument with a fact, countering them with another fact before suggesting a policy might be more convincing. Notably, this dialogic nature has been mostly overlooked in the computational analysis of argumentative discussions so far.

In this paper, we examine the nuances of ongoing argumentation in dialogues with the goal of understanding the particular elements that contribute to the success of particular debaters. Through a detailed analysis of ADU sequences, we seek to reveal the underlying patterns and strategies utilized by skilled debaters. These strategies are essential for advancing both theoretical understanding and practical application. Theoretically, they particularly improve our understanding of persuasive dialogues, enabling the refinement of skills among novice debaters and thereby enhancing the overall quality of discussions. In practical terms, these strategies serve as valuable guidance for the automated generation of persuasive texts that resonate effectively with various audiences. Moreover, a set of core strategies can potentially form the backbone of advanced writing tools, providing substantial support to writers as they structure their arguments.

To reveal these strategies, we introduce a new model for argument arrangement. This model includes identifying particular types of ADUs within a given argumentative discourse, analyzing the sequences of these ADUs, and clustering these sequences to reveal the patterns and strategies. We test our model using a large-scale dataset of persuasive discussions gathered from Change My View.[1] We sampled two types of discussion branches, dialogue and polylogue, and identified 16 clusters of similar ADU type patterns, each representing a different strategy used in the discussions. The sequence of clusters for each discussion (one cluster assignment per comment) then serves as a model of the discussion, which we evaluate against the task of determining the persuasiveness of the commenter.[2] For this task of persuasiveness detection, our model outperforms several strong baselines.

Overall, this paper introduces a new model for identifying argument arrangement strategies through clusters of similar ADU type patterns (Section 3). We develop a large-scale dataset comprising 34,393 discussion branches and completely tag

it for ADU types using a new approach that outperforms the previous state-of-the-art by 0.18 in terms of $F_1$ score (Section 4). Moreover, we use our model to identify clusters representing the arrangement strategies used in these discussions, and show the utility of cluster sequences for argumentation analysis through the example task of predicting persuasiveness (Section 5).[3]

## 2 Related Work

Argumentative discussions play a major role in computational argumentation analysis, for example when detecting counterarguments (Wachsmuth et al., 2018b) or whether two arguments are on the same side (Körner et al., 2021). A special emphasis is placed on the Change My View platform due to its extensive user base, strict moderation, and user-provided persuasiveness rating. Several prior works examined individual comments, where a wide range of linguistic, stylistic, and argumentative features have been employed to predict their persuasiveness (e.g., Tan et al. (2016); Hidey et al. (2017); Persing and Ng (2017); Morio et al. (2019)). For analyses beyond individual comments, Ji et al. (2018) employ features of both comments and the corresponding opening post, but do not look at discussions that are longer than one comment to the opening post. Guo et al. (2020) consider whole discussions and employ several linguistic features, but none tied to arrangement. Our paper makes a significant contribution to the task of predicting persuasiveness, going beyond exploring individual comments and delving into the arrangement of arguments within discussions.

**Argument models in online discussions** Hidey et al. (2017) model unit types for premises (ethos, logos, and pathos) and claims (interpretation, evaluation, agreement, or disagreement), revealing that the relative position of types can indicate a comment's persuasiveness. Morio et al. (2019), on the other hand do not distinguish between premises and claims. They use the following types: testimony, fact, value, policy, or rhetorical statements. We employ the types of Morio et al. (2019) for their simplicity, having shown impact on persuasiveness, and the easy to understand semantics behind the types (cf. Section 4.3), making them well-suited for modeling argument arrangement.

---

[1] https://www.reddit.com/r/changemyview
[2] We gauge persuasiveness according to the standards of the Change My View community guidelines, where the original poster signals that their view has been changed by marking comments with a $\Delta$ symbol in response.

[3] All the resources developed in the paper can be found on: https://github.com/webis-de/EMNLP-23

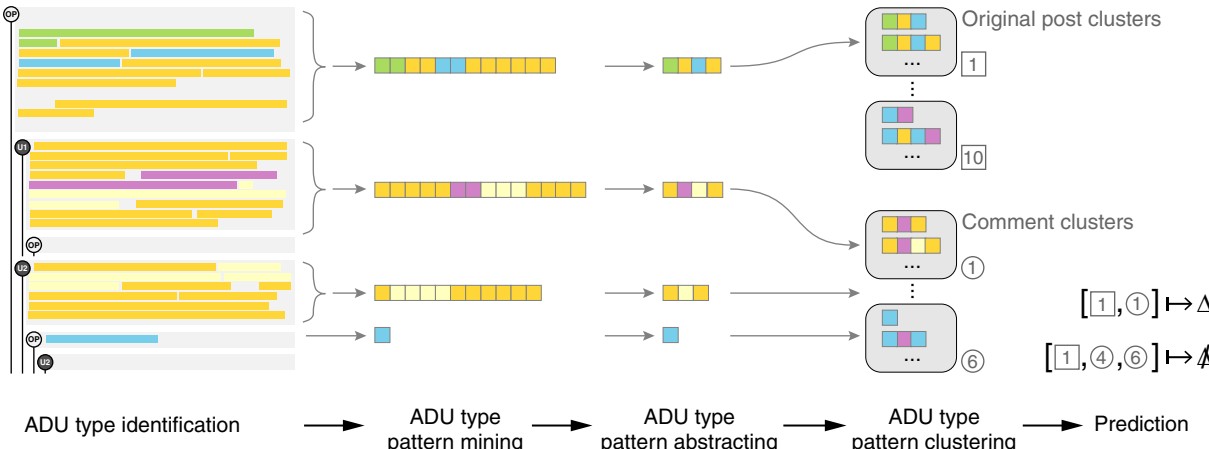

Figure 2: Pipeline for modeling the overall strategy as flows of ADU type arrangements per its opening post and comments. Illustrated on the two discussion branches of the example from Figure 1. The clusters are determined based on the abstracted patterns of all opening posts and comments in the dataset, respectively.

**Argument arrangement in online discussions**
The investigation of argument arrangement in social media platforms and its influence on persuasiveness remains a relatively understudied area with only a few works so far. Hidey et al. (2017) explored whether the types of their proposed model (see above) follow a specific order within Change My View discussions. They identified certain sequential patterns, such as the tendency for pathos premises to follow after claims of emotional evaluation. Morio et al. (2019) modeled persuasion strategies by the positional role of argumentative units, also by examining Change My View discussions. They find that facts and testimonies were commonly positioned at the beginning of posts, indicating the importance of presenting factual information before making claims. Conversely, policy suggestions tend to appear towards the end of posts, implying that recommendations or courses of action were treated as conclusions in the argumentation process. The closest work to ours is that by Hidey and McKeown (2018), who investigate the impact of argument arrangement on persuasiveness, albeit only within individual comments. They predict persuasiveness using word-level features, Penn Discourse TreeBank relations, and FrameNet semantic frames. In contrast, our work incorporates a clustering step that facilitates grouping similar sequences of argument unit types, enabling a more thorough exploration of arrangement strategies.

**Argument arrangement in other domains** A pioneering work of Al-Khatib et al. (2017) in the study of argument arrangement identified evidence patterns in 30,000 online news editorials from the New York Times and associated them with persuasive strategies. Through their analysis, they established specific rules for constructing effective editorials, like that arguments containing units of type testimony should precede those containing units of type statistics. Wachsmuth et al. (2018a) emphasized the importance of argument arrangement as a primary means in the context of generating arguments with a rhetorical strategy. They proposed and conducted a manual synthesis of argumentative texts that involves specific selection, phrasing, and arrangement of arguments following an effective rhetorical strategy. In contrast, our research specifically focuses on persuasive discussions. While there is an overlap in the argument unit types employed between our study and the work of Al-Khatib et al. (2017), the central focus in the former is on the concept of evidence. Moreover, our findings regarding argument arrangement hold the potential to be incorporated into the approach by Wachsmuth et al. (2018a) for generating persuasive arguments.

## 3 Modeling Argument Arrangement

Although various studies have explored the selection and phrasing of arguments in online discussions, often by analyzing the distribution of argument attributes and linguistic features (e.g., Wiegmann et al. (2022)), the effect of argument arrangement has received relatively limited attention. This paper aims to address this research gap by investigating the impact of argument arrangement, not only in predicting persuasiveness but also in gaining a deeper understanding of how individuals en-

gage in persuasive discussions.

In our approach to modeling argument arrangement in opening posts and comments, we employ a three-step pipeline, as illustrated in Figure 2. First, we identify the argumentative discourse units (ADUs) and their corresponding semantic types within the opening post and comments (Section 3.1). Next, we mine and abstract the sequence of ADU types to discover overarching patterns (Section 3.2). Finally, we categorize these patterns into clusters based on their similarity to other patterns (Section 3.3).

## 3.1 ADU Type Identification

For this step, we utilize the five ADU types introduced by Morio et al. (2019): fact, policy, testimony, value, and rhetorical question. One advantage of using these specific ADU types is the availability of a manually annotated dataset that serves as a valuable resource for fine-tuning a Large Language Model to effectively classify the ADU type in a given text. By identifying and categorizing these ADU types, we lay the foundation for understanding the structure and organization of arguments within the discussion.

## 3.2 ADU Type Pattern Mining

Once the ADU types are identified, we examine the sequences in which these types appear within the comments. To enhance the mining of more general and reliable sequences, we incorporate the "Change" transformation proposed by Wachsmuth et al. (2015). This transformation focuses on the transitions or changes between ADU types within a sequence. Through this abstraction, we emphasize the shifts in ADU types rather than the specific instances of each type. For example, if a sequence initially consists of [policy, policy, fact], it is abstracted to [policy, fact].

## 3.3 ADU Type Pattern Clustering

Once the patterns are identified, they are grouped into clusters. The assigned cluster's ID is then considered the sole representation of the arrangement of the opening post or comment. These clusters are identified using a clustering algorithm — we use hierarchical agglomerative clustering for its simplicity and adaptability — on all patterns of the dataset.

A critical element for pattern clustering is the measure of distance between patterns. Due to its importance for the clustering and thus for our approach, we propose and evaluate two measures:

**Edits** The Edits approach calculates the normalized distance between two sequences by quantifying the minimum number of edits (insertions, deletions, or substitutions) needed to transform one sequence into the other. We employ the widely used Levenshtein distance to determine the edit count. Yet, to ensure fairness, the resulting distance is normalized by the length of the longer sequence. This normalization ensures that an edit made to a shorter sequence has a more significant impact on the overall distance compared to an edit made to a longer sequence.

**SGT Embeddings** In the Embeddings approach, the ADU type sequences are transformed into a fixed-dimensional space using the Sequence Graph Transform technique proposed by Ranjan et al. (2022). This technique offers the capability to capture a range of dependencies without introducing additional computational complexity. The distance between two embedded sequences is then determined by employing a standard distance metric, such as Euclidean distance.

After the distance matrix calculation using either of the above-mentioned methods, hierarchical clustering can be applied. Initially, each sequence is considered an individual cluster. The algorithm then proceeds to merge the most similar clusters iteratively based on the distance between them. This process continues until all sequences are merged into a single cluster or until a predefined stopping criterion is met. The resulting hierarchical clustering tree, also known as a dendrogram, provides a visual representation of the clusters' hierarchical relationships. It allows us to observe the formation of subclusters and the overall clustering structure. An optimal number of clusters can be determined either by setting a threshold on the dendrogram or by employing other methods, such as the Elbow method (Thorndike, 1953).

After performing clustering, the arrangement is represented by the longest common sequence of ADU types within each cluster. This way we can gain insights into the prevailing ADU types and their sequential order, which represents the characteristic arrangement pattern within that particular cluster.

## 4 Preparing a Dataset of Argument Arrangements in Online Discussions

To evaluate the effectiveness of our argument arrangement model, we applied it to real-world online discussions data, specifically focusing on the task of predicting persuasiveness. We collected a dataset comprising 34,393 discussion branches, extracted from 1,246 threads sourced from the Change My View online discussion platform via the public Reddit API. These discussions adhere to a structured format enforced by the community, with persuasive discussions being identified by the presence of a $\Delta$ character, in accordance with community guidelines, which serves as our ground truth (Section 4.1). For our analysis, we categorize discussions into two scenarios, dialogue and polylogue (Section 4.2), and automatically identify Argumentative Discourse Units (ADUs) and their respective semantic types (Section 4.3). Subsequently, we offer initial insights into the use of ADU types through a statistical analysis of their positional distribution within a discussion branch (Section 4.4).

### 4.1 Structure of Change My View Discussions

Change My View provides a dynamic and inclusive platform that encourages users to engage in thoughtful discussions, with a strong emphasis on presenting well-reasoned arguments. The discussions are carefully moderated according to the community rules[4] ensuring the discussions maintain a high standard of quality and remain focused on the topic at hand.

In the Change My View platform, discussions, known as *threads*, begin when a user, referred to as the *original poster*, submits a text-based *opening post* expressing and justifying their viewpoint on any controversial topic. As implied by the name of the platform, each such submission is a challenge to other users, known as *commenters*, to change the original poster's view by submitting textual comments as replies. Both the original poster and commenters have the ability to contribute additional comments. The original poster may choose to add comments to further defend their justifications, while other commenters can provide counterarguments or rebuttals. This iterative commenting process can lead to long chains of discussion. For the purposes of this paper, we define each path from the opening post to a comment without any subsequent replies as a *branch*. These branches represent distinct trajectories within the discussion, highlighting the progression and development of the conversation from the opening post to subsequent comments. When the original poster perceives that a commenter has successfully influenced their viewpoint, even if only to a minor extent, they indicate this by posting a response that includes the $\Delta$ character. In line with prior research, we consider only those branches that contain a $\Delta$ in a comment from the original poster as persuasive.[5]

### 4.2 Grouping Discussions by Scenario

For an in-depth analysis of persuasiveness, we categorize the branches into two subsets based on different discussion scenarios:

**Dialogue** In this scenario, discussions unfold with a sequential exchange of arguments between the original poster and a single unique commenter. Here, both parties take turns presenting their viewpoints and counterarguments in a back-and-forth manner.

**Polylogue** This scenario encompasses discussion branches where multiple commenters engage with the original poster. These interactions involve multiple participants attempting to persuade the original poster, resulting in a dynamic, multi-party conversation.

This differentiation allows us to examine the effectiveness of persuasive discourse in different discussion scenarios and gain insights into the dynamics of persuasion. Table 1 provides key numbers for each scenario: dialogues more often contain a $\Delta$ and are on average shorter than polylogues. Moreover, out of the 1,246 total threads, nearly all contain at least one dialogue (1,236) and the majority contain at least one polylogue branch (1,058).

### 4.3 Automatic Identification of ADU Types

The fundamental building blocks of our approach (cf. Section 3.1) are the semantic ADU types presented by Morio et al. (2019).[6] The authors provide the following definitions and examples of the five ADU types:

---

[4]Change My View community rules: `https://www.reddit.com/r/changemyview/wiki/rules`

[5]To avoid disclosing the label, the comments awarding and confirming the $\Delta$ itself were excluded from our dataset.

[6]Mostly the same as those of Park et al. (2015).

| Scenario | # Threads | Branches | | | |
|---|---|---|---|---|---|
| | | # | | Avg. texts | |
| | | $\Delta$ | $\cancel{\Delta}$ | $\Delta$ | $\cancel{\Delta}$ |
| Dialogue | 1,236 | 2,618 | 14,860 | 2.3 | 2.9 |
| Polylogue | 1,058 | 940 | 16,140 | 5.4 | 8.6 |
| Both | 1,246 | 3,393 | 31,000 | 2.8 | 5.4 |

Table 1: Number of unique threads, branches (i.e., paths from opening post to a leaf comment), and average number of texts per branch (opening post and comments) in the dataset for each scenario and split between branches were a delta was awarded ($\Delta$) or not ($\cancel{\Delta}$).

**Testimony (T)** is an objective proposition related to the author's personal state or experience such as the following: *I do not have children.*

**Fact (F)** is a proposition describing objective facts that can be verified using objective evidence and therefore captures the evidential facts in persuasions: *Empire Theatres in Canada has a "Reel Babies" showing for certain movies.*

**Value (V)** is a proposition that refers to subjective value judgments without providing a statement on what should be done: *it is absolutely terrifying.*

**Policy (P)** offers a specific course of action to be taken or what should be done: *intelligent students should be able to see that.*

**Rhetorical Statement (R)** implicitly states the subjective value judgment by expressing figurative phrases, emotions, or rhetorical questions: *does it physically hurt men to be raped by women (as in PIV sex)?*

To identify the ADU types within the collected discussions, we fine-tuned a pre-trained ELECTRA-Large model (Clark et al., 2020) on the human-annotated dataset of Morio et al. (2019). The model was specifically trained for sequence labeling at the post level, enabling it to detect argumentative discourse units and assign corresponding ADU types. We evaluated the performance of our model on a separate hold-out test set from the dataset used by Morio et al. (2019). The model achieved an $F_1$ score of 0.62, surpassing the performance of the original authors' model by 0.18 and in our view is sufficiently reliable for large-scale analyses. Given that both our dataset and the one by Morio et al. (2019) were sourced from Change My View platform, it is reasonable to expect a similar level of performance when applied to our dataset.

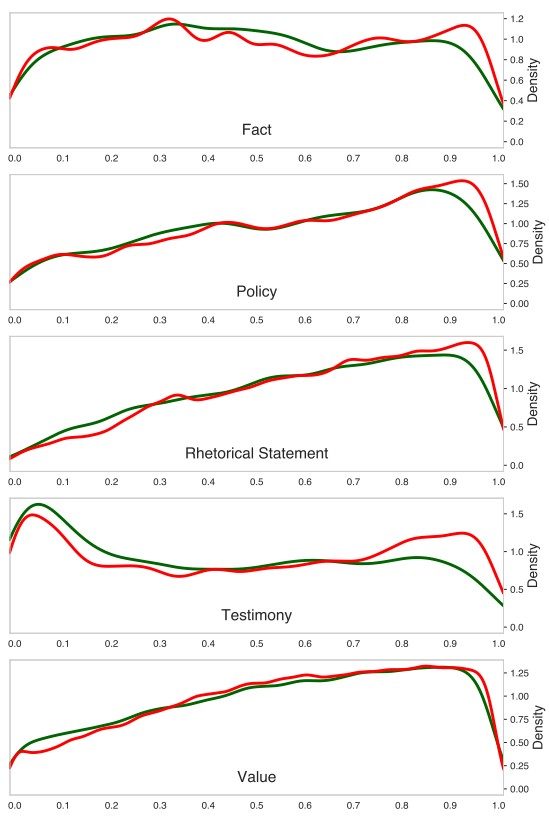

Figure 3: Distribution density of relative start positions (as a fraction of overall text length in a branch) for ADUs of each type, separately for branches with $\Delta$ (green) and without (red).

## 4.4 Positional Role of ADU Types

The analysis of the human-annotated dataset by Morio et al. (2019) revealed slight differences in the distribution of ADU types between persuasive and non-persuasive comments. They observed that testimonies and facts tend to occur more frequently at the beginning of texts, while policies are more prevalent towards the end. Figure 3 shows the result of reproducing their analysis on our larger dataset, utilizing automatically identified ADU types. As shown in the figure, our dataset exhibits a more balanced distribution of facts, whereas there is a noticeable increase in the usage of values and rhetorical statements towards the end of the discussions. We attribute these differences to our dataset containing not only the first comment to the opening post but complete discussion branches. Moreover, the figure suggests that even the position of ADU types alone can be a — though rather weak — predictor of persuasiveness. We anticipate that our exploration of ADU type arrangement will further enhance this predictive capability.

# 5 Analysis of Experiments and Results

In this section, we present the results of modeling argument arrangement of Argumentative Discourse Unit (ADU) types introduced in Section 3.3 and propose several approaches for predicting the persuasiveness of discussion branches. Furthermore, we conduct an experimental validation of our findings by incorporating cluster features in the task of persuasiveness prediction.

## 5.1 Identified Arrangement Clusters

Following our approach as described in Section 3.3, we cluster the ADU type patterns in our dataset. Notably, we conducted separate clustering for the patterns found in opening posts and comments, recognizing their distinct pragmatic purposes (justifying a view vs. arguing). To determine an appropriate number of clusters for each, we applied the widely-used Elbow criterion (Thorndike, 1953). Remarkably, this analysis revealed that both distance measures led to the same number of clusters for both opening posts (10 clusters) and comments (6 clusters).

Table 2 provides an overview of the identified clusters. Notably, when employing SGT embedding similarity, the clusters exhibit a more even distribution of texts. Directly related to that, the most frequent patterns per cluster tend to be shorter when employing SGT embeddings, especially when clustering comments. This indicates that in this approach short and frequent patterns are rather assigned to different clusters than concentrated in a single one. This approach holds promise for distinguishing various arrangement strategies effectively. Moreover, the table shows that the average length of patterns is quite different between clusters when using the Edit distance approach, which is also desirable for the same reason. Furthermore, we confirmed that the clusters also exhibit variations in the distribution density of ADU types, as demonstrated in Figure 3 for the entire dataset. The density plots for the clusters based on SGT embeddings can be found in the appendix (Figure 5).

## 5.2 Approaches for Predicting Persuasiveness

In order to evaluate the effectiveness of our argument arrangement model, we employ it in the context of predicting persuasiveness within discussions: Given a discussion branch, predict whether the original poster ultimately awards a Δ in the end. We compare the following approaches:

| Distance / Cluster | # Texts | Patterns | | |
|---|---|---|---|---|
| | | # | Avg. length | Most frequent |
| *Embeddings* | | | | |
| OP1 | 173 | 139 | 9.01 | VTVRV |
| OP2 | 159 | 96 | 7.29 | VFV |
| OP3 | 194 | 75 | 5.52 | TV |
| OP4 | 99 | 90 | 9.72 | VFVRV |
| OP5 | 74 | 68 | 10.26 | VPVFV |
| OP6 | 141 | 33 | 4.43 | VRV |
| OP7 | 134 | 38 | 4.61 | VPV |
| OP8 | 102 | 90 | 8.52 | VRVPV |
| OP9 | 53 | 43 | 8.55 | VTVPV |
| OP10 | 117 | 1 | 1.00 | V |
| C1 | 46,085 | 595 | 1.41 | VTV |
| C2 | 13,758 | 765 | 4.60 | VRV |
| C3 | 11,740 | 851 | 3.52 | FVR |
| C4 | 9,013 | 622 | 3.05 | TVR |
| C5 | 3,120 | 331 | 4.30 | VPV |
| C6 | 1,767 | 408 | 4.52 | PV |
| *Edits* | | | | |
| OP1 | 390 | 278 | 8.82 | VTV |
| OP2 | 106 | 105 | 16.30 | VFVFVFVFV |
| OP3 | 138 | 67 | 5.71 | VRV |
| OP4 | 202 | 35 | 2.01 | TV |
| OP5 | 79 | 28 | 3.82 | VPV |
| OP6 | 129 | 82 | 7.30 | VFV |
| OP7 | 46 | 24 | 5.12 | TVTV |
| OP8 | 19 | 19 | 8.15 | VRVF |
| OP9 | 25 | 25 | 10.42 | VFVTVR |
| OP10 | 112 | 76 | 8.33 | PVVPV |
| C1 | 72,702 | 1,957 | 2.20 | RV |
| C2 | 3,521 | 801 | 8.22 | VPVPVP |
| C3 | 1,008 | 316 | 7.46 | VFVFVF |
| C4 | 151 | 12 | 4.10 | FVRV |
| C5 | 1,742 | 342 | 6.31 | VFVFV |
| C6 | 2,869 | 144 | 3.78 | VTV |

Table 2: Overview of the identified clusters in our dataset, either via the embeddings or edits distance, and either in the opening posts (OP$X$) or comments (C$X$): number of texts (opening posts or comments) they contain as well as the number of unique patterns, their average length, and the most frequent one.

**Length-Based Classifier** As observed by Tan et al. (2016) and reflected in our dataset (cf. Table 1), lengthy discussions on Change My View tend to receive fewer Δ awards. To establish a preliminary baseline, we employ logistic regression to classify discussion branches as persuasive or not based solely on their length feature.

**Interplay Features** In the study of Tan et al. (2016), the most effective features for the persuasiveness prediction were extracted from the interaction between the opening post and the replies. They derived 12 features from 4 similarity measures (common words, similar fraction in response, similar fraction in the opening post, and Jaccard score) and 3 subsets (all words, stop words, and

| Test set / Model | Scenario | | | | | | | | | | | |
|---|---|---|---|---|---|---|---|---|---|---|---|---|
| | **Dialogue** | | | | **Polylogue** | | | | **Both scenarios** | | | |
| | $F_1$ | $F_{1\Delta}$ | $P_\Delta$ | $R_\Delta$ | $F_1$ | $F_{1\Delta}$ | $P_\Delta$ | $R_\Delta$ | $F_1$ | $F_{1\Delta}$ | $P_\Delta$ | $R_\Delta$ |
| *Unbalanced test set* | | | | | | | | | | | | |
| Length-Based Classifier | 0.56 | 0.37 | 0.25 | 0.75 | 0.49 | 0.19 | 0.11 | 0.71 | 0.52 | 0.30 | 0.18 | **0.81** |
| Interplay features | 0.54 | 0.47 | 0.37 | 0.65 | 0.48 | 0.15 | 0.09 | 0.52 | 0.53 | 0.33 | 0.22 | 0.68 |
| BERT | 0.55 | 0.21 | 0.52 | 0.13 | 0.48 | 0.24 | 0.50 | 0.14 | 0.59 | 0.25 | 0.47 | 0.17 |
| Edits | 0.78 | 0.66 | 0.59 | 0.74 | 0.74 | 0.52 | 0.48 | 0.53 | 0.61 | 0.40 | 0.28 | 0.74 |
| SGT Embeddings | 0.82 | 0.71 | 0.66 | 0.77 | **0.75** | **0.53** | **0.53** | 0.56 | 0.61 | 0.42 | 0.28 | **0.81** |
| BERT+Edits | 0.80 | 0.68 | 0.60 | 0.77 | 0.72 | 0.47 | 0.45 | 0.49 | 0.77 | 0.58 | 0.57 | 0.58 |
| BERT+SGT Embeddings | **0.83** | **0.72** | **0.67** | **0.81** | **0.75** | 0.51 | **0.53** | 0.56 | **0.78** | **0.60** | **0.62** | 0.58 |
| Llama-7B Zero-Shot | 0.24 | 0.18 | 0.12 | 0.44 | 0.23 | 0.17 | 0.11 | 0.41 | 0.23 | 0.16 | 0.10 | 0.34 |
| Llama-7B Branches | 0.32 | 0.27 | 0.18 | 0.71 | 0.30 | 0.24 | 0.15 | 0.65 | 0.29 | 0.25 | 0.16 | 0.65 |
| Llama-7B Branches+Strategies | 0.36 | 0.33 | 0.21 | 0.74 | 0.36 | 0.32 | 0.20 | **0.73** | 0.37 | 0.34 | 0.22 | 0.76 |
| *Balanced test set* | | | | | | | | | | | | |
| Length-Based Classifier | 0.64 | 0.68 | 0.63 | 0.75 | 0.62 | 0.65 | 0.61 | **0.70** | 0.56 | 0.69 | 0.57 | **0.86** |
| Interplay features | 0.54 | 0.45 | 0.47 | 0.43 | 0.59 | 0.51 | 0.64 | 0.43 | 0.53 | 0.50 | 0.47 | 0.53 |
| BERT | 0.73 | 0.76 | 0.70 | 0.84 | 0.60 | 0.60 | 0.54 | 0.67 | 0.64 | 0.70 | 0.62 | 0.80 |
| Edits | 0.81 | 0.80 | 0.86 | 0.74 | 0.70 | 0.65 | 0.78 | 0.56 | 0.71 | 0.71 | 0.70 | 0.72 |
| SGT Embeddings | 0.84 | **0.84** | **0.90** | **0.78** | **0.75** | **0.71** | **0.87** | 0.60 | 0.72 | **0.73** | 0.70 | 0.77 |
| BERT+Edits | 0.80 | 0.68 | 0.61 | 0.77 | 0.68 | 0.61 | 0.79 | 0.50 | 0.70 | 0.69 | 0.70 | 0.56 |
| BERT+SGT Embeddings | **0.85** | 0.83 | **0.90** | **0.78** | 0.72 | 0.68 | **0.87** | 0.49 | **0.75** | 0.70 | **0.90** | 0.70 |
| Llama-7B Zero-Shot | 0.31 | 0.46 | 0.55 | 0.40 | 0.27 | 0.38 | 0.46 | 0.32 | 0.29 | 0.40 | 0.47 | 0.35 |
| Llama-7B Branches | 0.40 | 0.62 | 0.58 | 0.66 | 0.39 | 0.60 | 0.60 | 0.61 | 0.37 | 0.60 | 0.62 | 0.59 |
| Llama-7B Branches+Strategies | 0.45 | 0.64 | 0.67 | 0.62 | 0.43 | 0.66 | 0.64 | 0.69 | 0.45 | 0.67 | 0.66 | 0.68 |

Table 3: Effectiveness of the tested models in predicting whether a discussion branch received a $\Delta$ in the different scenarios and for both the unbalanced (full) and a balanced test set. Employed measures are Macro-$F_1$ score ($F_1$) as well as the $F_1$ score, precision, and recall for detecting delta discussions ($F_{1\Delta}$, $P_\Delta$, $R_\Delta$).

content words). As a lexical baseline approach, we employ the same features and use logistic regression to classify the discussion branches.

**BERT-Based Classifier** As a baseline that captures both the semantic and syntactic information of the text, we employ a classifier based on BERT$_{large}$ embeddings (Devlin et al., 2019). For this approach, we concatenate the BERT embeddings for each turn in the discussion and pass them through a bidirectional LSTM (Hochreiter and Schmidhuber, 1997) to capture contextual dependencies between different turns. The LSTM's outputs are then processed through a linear layer and softmax activation to predict persuasiveness.

**Argument Arrangement: Clusters** In this approach, we utilize a bidirectional LSTM model in conjunction with a linear layer and softmax activation for predicting persuasiveness labels based on the identified cluster features. The input to the LSTM consists of a sequence of cluster features identified by either the SGT embeddings or Edit distance approach.

**Combination of Contextualized Features and Argument Arrangement** The study by Li et al. (2020) demonstrated that incorporating argument structure features into an LSTM model plays an

essential role in predicting which round of debate (Pro vs. Con) makes a more convincing argument on the debate.org website. In a similar manner, to leverage the strengths of both the ADU type arrangements and BERT embeddings, we combine the outputs from the previous two models. Two linear layers with softmax are used to predict the output probabilities over both of these LSTM models separately.

**Llama-2-7B Zero-Shot** We utilize the out-of-the-box Llama-2 model with seven billion parameters (Touvron et al., 2023). Given the discussion branch, the model was prompted to generate binary answers (yes/no) when asked whether the original poster's opinion was altered by the end of the discussion.

**Llama-2-7B Fine-Tuned Branches** Using the same prompt as in the zero-shot approach, we fine-tune the Llama-2-7B model on the discussion branches, with the true yes/no answers provided in the prompt. We implement low-rank adaptation (Hu et al., 2022) with parameters $r = 64$ and scaling factor $\alpha = 16$ applied to all linear layers. The model was fine-tuned for one epoch, as further training did not yield significant reduction in the loss, with a batch size of 8 and a learning rate of $2 \times 10^{-4}$ on a single NVIDIA A100 GPU (40GB).

**Llama-2-7B Fine-Tuned Branches+SGT Clusters** This approach closely mirrors the one above, with a slight variation in the prompt. Here, we extend the prompt with cluster features identified from the dataset using the SGT Embeddings approach. Figure 4 illustrates the prompt template used in all three configurations.

## 5.3 Results of Predicting Persuasiveness

We present the results of our persuasiveness prediction analysis for all branches under the two scenarios described in Section 4.2. Considering the class imbalance, we evaluate the approaches on the complete dataset as well as a balanced sample to address the imbalance issue. For the evaluation, we divide the branches into training and test sets, maintaining a ratio of $8 : 2$. The predictions are then assessed on the held-out test set using the macro-$F_1$ measure. Given a heavily skewed label distribution, we want to emphasize the importance of $\Delta$ prediction, thus we provide more detailed evaluation results for this label. The effectiveness of the classifiers is reported in Table 3.

The classification results reveal several findings: First, the length baseline provides a starting point for comparison of the discussion branches in the balanced setting, achieving an overall $F_1$ score of 0.56 in both scenarios and 0.69 $F_1$ score for the $\Delta$ discussions. While the length baseline provides a basic understanding of the relationship between discussion length and persuasiveness, we recognize the need for more sophisticated methods to capture the nuances of persuasive discourse. Second, our experiments with encoding the discussions using BERT embeddings have yielded promising results, however, we have observed that incorporating argument arrangement features further enhances the prediction performance. In fact, solely using cluster features to represent the discussion dynamics is effective for persuasion prediction. We can see a similar trend with the Llama-2-7B-based approaches, where the performance of the model significantly improves after fine-tuning on the discussion branches, but even more so when incorporating the identified strategy features in the prompt.

The enhanced prediction performance achieved by incorporating argument arrangement features highlights their significance in capturing the intricate dynamics of persuasive discussions.

```
### Instruction:
Below is a conversation between OP and one or more
users along with the persuasion strategies employed
by each of them. Read the discussion and decide
whether by the end of the discussion the opinion of
the OP was changed.
### OP (used strategy X): <opening post text>
### Reply 1 (used strategy Y): <comment 1>
...
### Reply N (used strategy Z): <comment N>
### The opinion of the OP was changed [yes/no]:
<model response>
```

Figure 4: Example of a prompt we employ for zero-shot and fine-tuning experiments. The text colored in blue is adapted to the conversation at hand. The text colored in red is used only when identified clusters are employed.

## 6 Conclusion

This paper expanded our understanding of online persuasive dialogues by uncovering that debaters follow certain arrangement strategies. We introduced a new model for arrangement strategies that is based on patterns of argumentative discourse unit types. Clusters of such patterns correspond to different strategies, with sequences of these clusters (one element per comment of a discussion branch) representing a whole discussion. This model was operationalized using a large-scale dataset comprising 34,393 discussion branches. In a comparative evaluation of ten approaches, we demonstrate the remarkable utility of these arrangement strategies in predicting persuasiveness — both if used as sole feature and in addition to others —, emphasizing their essential role in unraveling the dynamics of persuasive online discussions.

Still, there's ample room for refining and expanding this research further. One aspect worth exploring is the development of more fine-grained categories for argumentative discourse units, such as incorporating the human value categories proposed by Kiesel et al. (2022). Besides, the identified arrangement strategies have applications beyond persuasiveness prediction. Exploring their potential in tasks such as writing assistance and text generation could broaden the scope of argumentative discourse analysis into new horizons.

## 7 Limitations

While our research contributes valuable insights into the prediction of persuasiveness and the role of argument arrangement, it is important to acknowledge certain limitations that may impact the generalizability and interpretation of our findings.

**ADU Type Classification** We rely on automated methods for the identification and classification of argumentative discourse units (ADUs), which may introduce errors or inaccuracies. Despite efforts to ensure accuracy, misclassifications or inconsistencies in ADU labeling could potentially impact the analysis and predictions. Further refinement and validation of the ADU classification process are necessary for more robust results.

**Limited Scope of Features** Our study primarily focuses on ADU types and argument arrangement features in predicting persuasiveness. While these features have shown promising results, there are certainly other important linguistic, contextual, or stylistic features that were not considered in our analysis. Future research should explore combinations of such features with arrangement features, like we did with BERT embeddings, to better understand their individual and combined impact on persuasiveness prediction.

**Only One Platform** Our study focuses primarily on the Change My View platform, which may limit the generalizability of our findings to other social media platforms or contexts. The characteristics and dynamics of persuasive discourse may vary across different platforms, user demographics, and topics. Future research should explore the generalizability of our findings to a broader range of platforms and contexts.

## 8 Ethics Statement

In terms of data collection, this study utilizes publicly available datasets that strictly adhere to ethical considerations, ensuring compliance with Change My View policies and maintaining the anonymity of the users who participated in the discussion.

As for our research findings, they have implications for the identification and potential generation of persuasive text, especially within the limitations mentioned above. Hence, it is important to recognize the potential misuse of this capability, as malicious actors may in the future be able to employ our approach as one one many building blocks to rapidly create misleading or deceptive text for a target audience with a profound persuasive impact, for example in chatbots.

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

# A Appendix

For reference, Figure 5 provides the density plots for the clusters based on embeddings. Figure 6 provides the density plots for the different ADU types in replies based on OP cluster and whether a $\Delta$ was awarded or not.

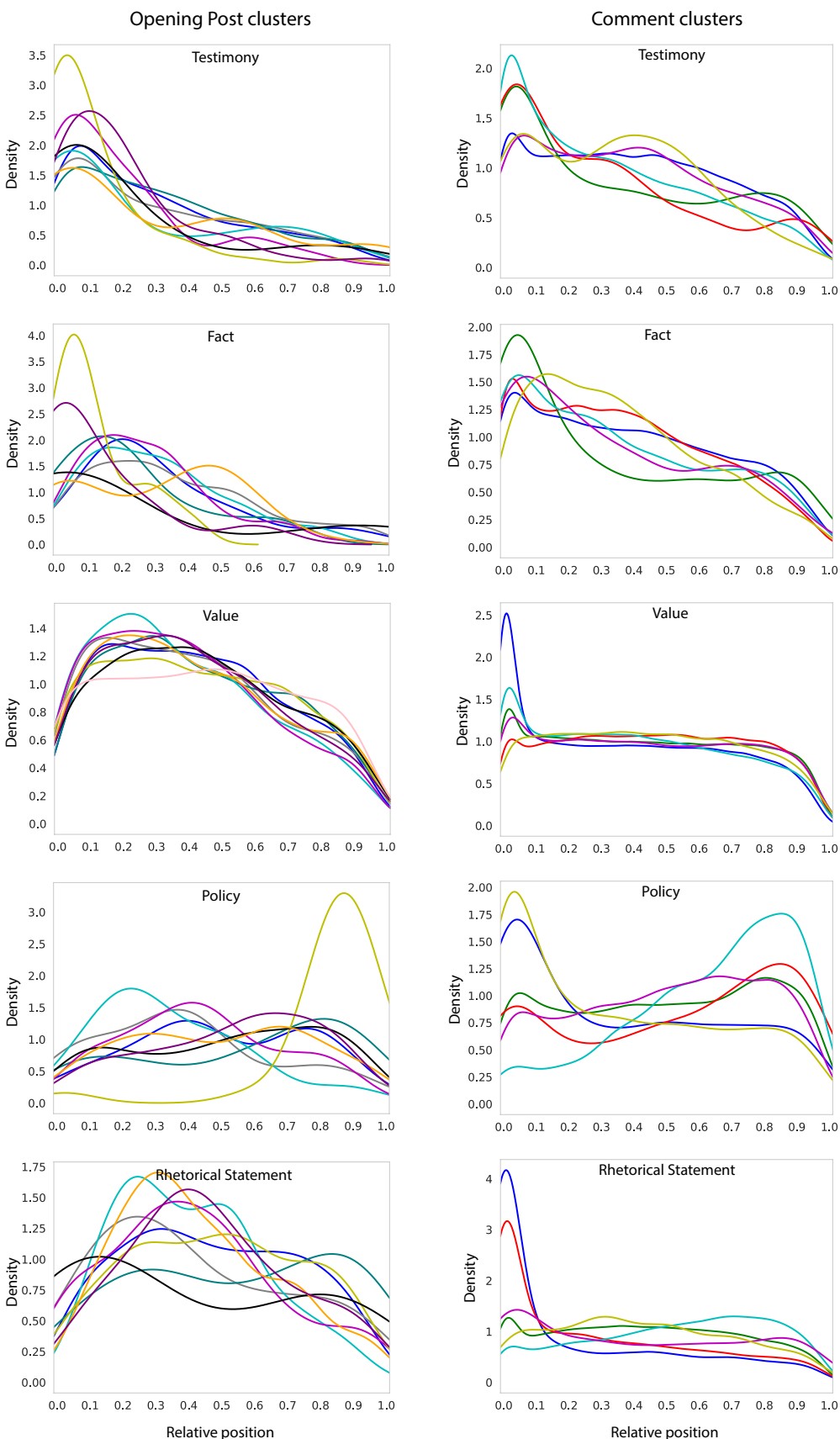

Figure 5: Distribution density of relative start positions (as fraction of overall text length) for ADUs of each ADU type, separately for the different embedding-based clusters (color-coded) of opening posts (left) and comments (right). The density graphs illustrate how the patterns in some cluster use certain ADU types rather at the start and others types rather at the end of the texts.

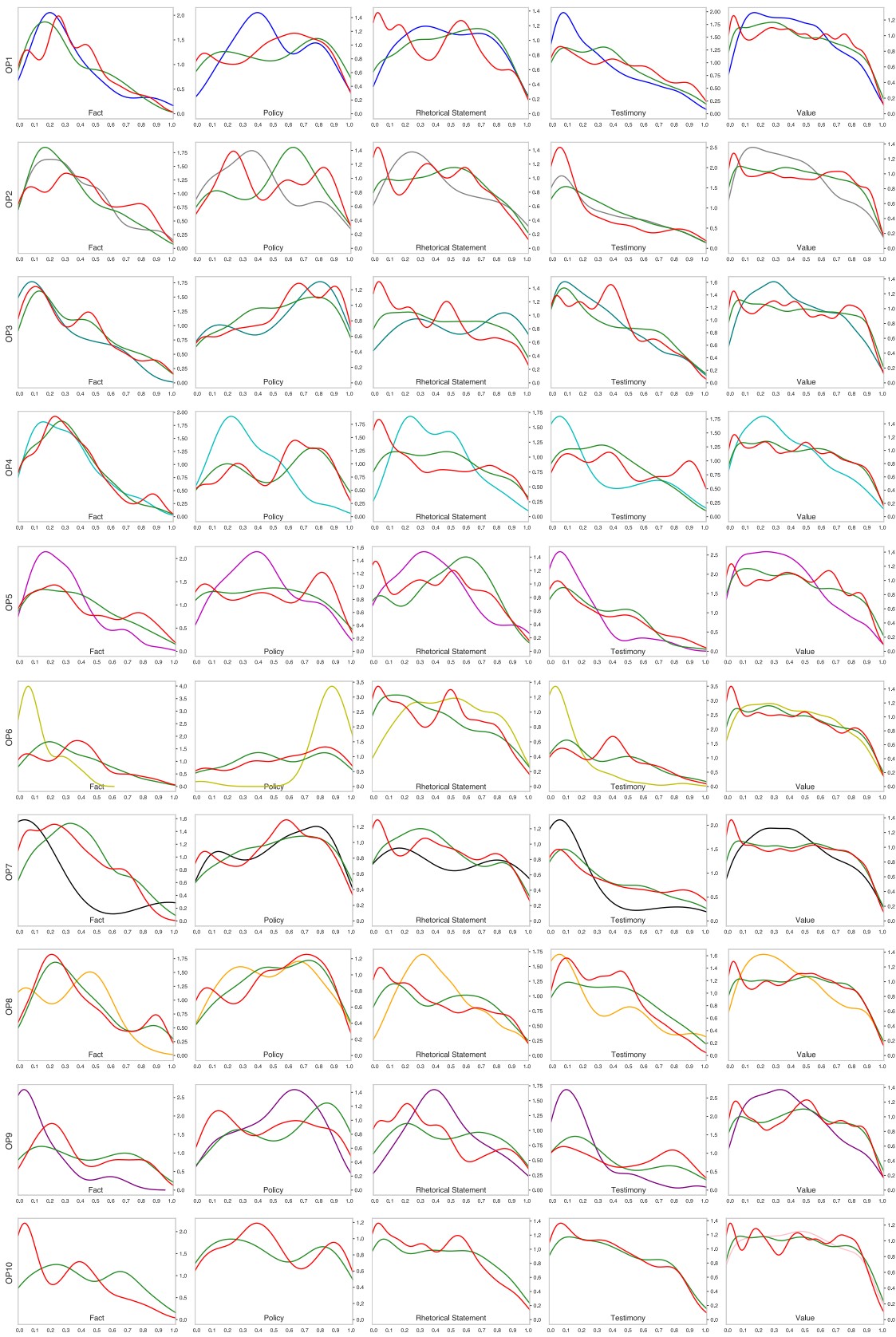

Figure 6: Distribution density of relative start positions, expressed as a fraction of the overall text length, for each ADU type. Each row represents the distribution within one of the ten identified original poster (OP) embedding-based clusters (color-coded), and the corresponding reply branches, distinguished by whether they were awarded a Δ (green) or not (red). The density graph reveals similarities and differences in the usage of certain ADU types across commenters in response to different OP strategies.