# OpenReview forum: "Unveiling the Power of Argument Arrangement in Online Persuasive Discussions"
_EMNLP/2023/Conference — EMNLP 2023 Findings_

### Official Review · Reviewer_HDqE · 2023-08-02

**Soundness:** 3

**Excitement:**

2: Mediocre: This paper makes marginal contributions (vs non-contemporaneous work), so I would rather not see it in the conference.

**Missing References:**

Ji et.al., 2018. Incorporating Argument-Level Interactions for Persuasion Comments Evaluation using Co-attention Model. COLING.
Guo et.al., 2020. In Opinion Holders’ Shoes: Modeling Cumulative Influence for View Change in Online Argumentation. 2388-2399. WWW.



**Paper Topic And Main Contributions:**

The paper studies the impact of argument pattern on persuasiveness in online discussion. The authors claim that existing research mostly focuses on individual comments and ignore the interactive relationship between arguments. They propose to use ADU (argumentative discourse unit) patterns to capture the structure of discussions. They fine-tune a pre-trained LLM to classify ADU type and propose two approaches, Edits and embeddings based, to cluster ADUs. The authors analyze the data they collected to understand the ADU type distribution and cluster statistics. To verify the effectiveness of using argument patterns for predicting persuasiveness, the authors incorporate the argument structure features into an LSTM model and compare the model with length based and BERT based baselines.



**Questions For The Authors:**

What is the benefit of creating a new CMV dataset?



**Reasons To Accept:**

- The paper studies an interesting problem.
- The proposed method of capturing argument structure makes sense

**Reasons To Reject:**

The interactive relationship between arguments have been studied before. The paper does not provide comparison with existing approaches.

The paper leverages LLM for ADU identification. However, the quality verification is rather ad-hoc. The authors verify their model using a different CMV data and claim the two datasets are similar. (1) CMV contains topics from a large variety of domains. (2) If the two datasets are similar, what is the benefit of creating a new dataset?



**Reproducibility:**

4: Could mostly reproduce the results, but there may be some variation because of sample variance or minor variations in their interpretation of the protocol or method.

**Reviewer Confidence:**

5: Positive that my evaluation is correct. I read the paper very carefully and I am very familiar with related work.

---

> ### Author Rebuttal · Authors · 2023-08-29
>
> We appreciate the reviewer's thorough evaluation of our paper on the impact of argument arrangement patterns on persuasiveness in online discussions. We address the points raised in the review below:
>
> We thank the reviewer for acknowledging the significance of the problem we addressed. The impact of argument structure on persuasiveness is indeed an interesting area of research. We appreciate the acknowledgment of the sensibility of our ADU-pattern-based approach. Capturing the structure of the discussions through ADUs indeed provides a holistic understanding of the persuasion dynamics.
>
> __Response to Reasons to Reject and Missing References:__
>
> - __For the existence of studies on interactive relationships between arguments:__
> While some research touches on this aspect, we emphasize that the majority either neglected the truly interactive nature of arguments or focused on a simplistic dialogical structure involving a single OP post and one reply, which was either awarded a delta or not, which in turn raises the question of how these replies have been selected: was the delta awarded directly to this reply or after several back-and-forth turns? We address these concerns by investigating more complex levels of interactions, collecting a new dataset with an enhanced structure, and more up-to-date discussions. Our paper is novel in its deep examination of the intricate argumentative patterns within online discussions.
>
> - __For the additional references of Ji et al. and Guo et al.:__
> While these studies do touch upon the dialogical nature of CMV discussions, the work of Ji et al. primarily focuses on simple dyadic interactions (OP post and a single reply), not capturing the complex dialogue dynamics we address in our study. Guo et al, in turn, do consider whole discussions, but they target a different task of predicting the points of opinion change on every turn in a discussion. On the other hand, our method delves into multi-turn discussions and employed argumentation strategies. We will add the two studies to our related work section and discuss the difference between their approach and ours.
>
> - __For the lack of direct comparisons with existing approaches:__
> We want to clarify that our approach is fundamentally distinct, as it delves into strategies employed in multi-turn dialogues, an area that has remained largely unexplored. The application of including these strategies as features for persuasiveness prediction is just one manifestation of the broader potential our approach holds, extending beyond this specific task. However, our clusters can indeed be combined with existing persuasiveness models. As a concrete example, we now conducted some additional experiments, involving the approach proposed by Tan et al (2016) and in particular using the best performing features: ‘Interplay’. In terms of macro F1 scores, adding cluster features to the set increases the performance from 0.53 to 0.68 for root paths and from 0.51 to 0.58 for the whole paths. As for the ROC-AUC score, we get an increase from 0.62 (similar to what Tan et al report) to 0.75 for whole paths and from 0.57 to 0.67 for the root paths, respectively.
>
> - __Leveraging an LLM for ADU Identification:__
> Indeed, our paper employs a pre-trained ELECTRA-1.75M model, fine-tuned for ADU identification. While the reviewer characterizes the quality verification as "ad-hoc," we would like to clarify that our model's performance evaluation was rigorously conducted using a separate hold-out test set from the dataset of Morio et al. (2019). The model achieved a macro F1 score of 0.62, surpassing the original authors' model performance by 0.18. This evaluation encompasses a standard practice to validate the effectiveness of the model.
>
> - __The value of creating a new CMV dataset:__
> While we acknowledge the diversity of the initial CMV dataset, we would argue that CMV datasets are still quite similar to each other in terms of language and discussion structure (the latter being enforced throught moderation). However, we advocate for the creation of an updated version due to the time gap since its release in 2016, as was pointed out by the reviewer [w6PD](https://openreview.net/forum?id=0eWQVWvPgu&noteId=9k3fvnsX0B). Moreover, another of the new dataset's advantages lies in its improved structure: threads in the new dataset are organized into dialogue branches, which enhances our ability to capture and analyze the interactive nature of argument patterns more effectively. This structure is crucial for our proposed approach, which aims to uncover the nuanced strategies underlying persuasive interactions within discussions.
>
> __Response to Questions for The Authors:__
>
> Please see our answer to the same question in the Response to Reasons to Reject and Missing References.
>
> Thank you for your thoughtful review, and we look forward to any further feedback or inquiries you may have.

---

### Official Review · Reviewer_Sf3y · 2023-08-05

**Soundness:** 3

**Excitement:**

2: Mediocre: This paper makes marginal contributions (vs non-contemporaneous work), so I would rather not see it in the conference.

**Paper Topic And Main Contributions:**

 The authors study the argumentation in online discussions by analyzing the argument arrangement structures and finding common patterns representative of argumentation strategies used in online discussions. In particular,

1. authors identify arrangement strategies based on clusters of similar ADU type patterns using pattern clustering methods (edits and embeddings based)
2. provide a new CMV based dataset to study persuasiveness
3. show that these identified patterns are predictive of persuasiveness.

**Questions For The Authors:**

 1. Frequent pattern mining algorithms could be an alternative to identifying such patterns. Did you experiment with patterns with exact sequences of ADUs, instead of the current approach and what were the potential limitations of such an approach?


**Reasons To Accept:**

 1. The method of identifying structure of ADUs based on pattern mining and pattern clustering is well motivated and supported by experiments.
2. Analysis of roles of ADU types is interesting and can be further emphasized.
3. The identified patterns as features for persuasiveness prediction provides an interpretable approach.


**Reasons To Reject:**

 1. Paper can explore more recent models for persuasiveness prediction. It would further strengthen the paper if more recent models also benefit from including such argument structure patterns as features.

**Reproducibility:**

3: Could reproduce the results with some difficulty. The settings of parameters are underspecified or subjectively determined; the training/evaluation data are not widely available.

**Reviewer Confidence:**

2: Willing to defend my evaluation, but it is fairly likely that I missed some details, didn't understand some central points, or can't be sure about the novelty of the work.

---

> ### Author Rebuttal · Authors · 2023-08-29
>
> We appreciate the thoughtful assessment provided by the reviewer regarding our paper, which delves into the analysis of argument arrangements in online persuasive discussions. We would like to address the points raised by the reviewer.
>
> First and foremost, we are grateful for the recognition of our paper's main contributions. The identification of argument arrangement strategies based on clusters of similar Argumentative Discourse Unit (ADU) type patterns is indeed a pivotal aspect of our work. We appreciate that the reviewer acknowledges the novelty, significance, and interpretability of our approach.
>
> __Response to Reasons to Reject:__
>
> For employing more recent models for persuasiveness prediction, we acknowledge the importance of staying up-to-date with the latest advancements in the field. However, our paper's main focus is not on proposing a new persuasiveness prediction model but rather on analyzing the argument arrangement structures and identifying common patterns in online discussions. While future research could explore more recent models, it doesn't detract from the contributions we present in our current work. However, during this short rebuttal period, we were able to conduct some additional experiments involving the approach proposed by Tan et al (2016) and in particular using the best performing features: ‘Interplay’. In terms of macro F1 scores, adding cluster features to the set increases the performance from 0.53 to 0.68 for root paths and from 0.51 to 0.58 for the whole paths. As for the ROC-AUC score, we get an increase from 0.62 (similar to what Tan et al report) to 0.75 for whole paths and from 0.57 to 0.67 for the root paths, respectively. These results show that also other models benefit from including such argument structure patterns as features.
>
> __Response to Questions for the Authors:__
>
> - __Frequent Pattern Mining Algorithms:__
> We appreciate the reviewer’s suggestion to use Frequent Pattern Mining as it highlights a relevant approach. Such algorithms would be more suited for scenarios where the goal would be to discover the exact frequent sequential patterns within the data. However, in our paper, the goal is to analyze how argument arrangements can be grouped based on their sequential similarity. Thus, we found clustering to be more appropriate, as it allows us to explicitly specify the similarity measure.
>
> - __Experiments with patterns with exact sequences of ADUs:__
> We experimented with both exact sequences of ADUs and abstracted versions of sequences in our research. While considering exact sequences might seem intuitive, we encountered several limitations with this approach. First, the sheer diversity of discussions makes it challenging to capture precise patterns that account for all possible variations. Additionally, the abstraction through the removal of repetitions allows for greater generalizability across diverse discussions, enhancing the applicability of our findings. Our observations here align with those of related work, for example, the results of Wachsmuth et. al. (2015) in the domain of sentiment analysis.
>
> We're grateful for your insightful review and are looking forward to any additional feedback or inquiries you would like to share.

---

### Official Review · Reviewer_w6PD · 2023-08-05

**Soundness:** 3

**Excitement:**

3: Ambivalent: It has merits (e.g., it reports state-of-the-art results, the idea is nice), but there are key weaknesses (e.g., it describes incremental work), and it can significantly benefit from another round of revision. However, I won't object to accepting it if my co-reviewers champion it.

**Paper Topic And Main Contributions:**



The contributions are explicity:
- They release a dataset for reddit's Changemyview
- They use the ADUs from Morio et al 2019 and argue that the order/structure of these ADUs matter
- They identify two ways to operationalize this structure: edits and embeddings
- They evaluate the utility of these structures by evaluating their predictive power on whether or not a user changes their mind given a branch.

**Questions For The Authors:**

Question A: Did you happen to have some metric for cluster quality?


Question B: Or, if not, do you have a rationale for why the clusters for embeddings and edits have very uneven sized clusters, but the clusters seem to be very different? e.g., the most frequent in Edit C1 is RV, but the most frequent in Embedding C1 is VTV.

Question C: For Fig 3, all the green seems to taper off at the end. Do you happen to know what is happening? e.g., could they be artifacts of the last comments being the delta given?

**Reasons To Accept:**

- a newer dataset for CMV is interesting, especially given that Tan et al is from 2016
  - Section 4.4 on the analysis of the dataset with ADUs is quite interesting, although not central to the paper
- Idea for using patterns of ADU is novel and can be interpretable as a feature
- Innovations (Edit and Embeddings features) do seem to help a lot in their quantitative experiments

**Reasons To Reject:**

- Models used are fairly weak (BERT) compared to what we have available.
   - This could lead to the paper having less visibility
   - The paper validates the edit and embeddings features with predictive performance, so this may be in the reader's mind as they read the paper


**Reproducibility:**

4: Could mostly reproduce the results, but there may be some variation because of sample variance or minor variations in their interpretation of the protocol or method.

**Reviewer Confidence:**

2: Willing to defend my evaluation, but it is fairly likely that I missed some details, didn't understand some central points, or can't be sure about the novelty of the work.

**Typos Grammar Style And Presentation Improvements:**

I'd emphasize the usefulness of a new dataset for CMV. I think a very easy concern is "why duplicate? However, there has been almost 10 years of content since then.

---

> ### Author Rebuttal · Authors · 2023-08-29
>
> We appreciate the reviewer’s careful consideration of our paper and their insightful feedback. We address the raised concerns below and provide additional context for our work.
>
> In regard to the strengths of our work, we appreciate the reviewer’s acknowledgment of the novelty in releasing a new CMV dataset. We are glad that the reviewer finds our approach novel and points out its interpretability.
>
> __Response to Reasons to Reject:__
>
> We understand the reviewer’s concern about BERT being no longer considered state-of-the-art, but we chose it due to its reliability and simplicity. In this regard, it is crucial to note that the paper’s focus is not to achieve state-of-the-art results in predicting persuasiveness, but to analyze successful arrangement strategies in discussions. The persuasiveness classification is just a validation process that works well with BERT and can work (maybe even better) with more advanced models.
>
> __Response to Questions for the Authors:__
>
> - __Question A:__ Due to the unlabeled nature of our data, we faced limitations in assessing quality metrics for the resulting clusters beyond the internal evaluation measures we used to select the number of clusters via the Elbow criterion. Nevertheless, we conducted a thorough manual validation of the outcomes to guarantee the coherence, relevance, and absence of excessive overlap within the clusters. We will include this clarification in the updated version of the paper.
> - __Question B:__  Based on manual inspections, we hypothesize that the varying cluster sizes between the Embeddings and Edits method can be attributed to Edits method being less able to capture the inherent sequential similarities in ADU patterns. We'll make sure to highlight our hypothesis in the revised version.
> - __Question C:__ Indeed, one could wonder whether the text that awards the delta would have an influence on our results. However, this is not the case, because, to prevent the inference of delta prediction solely from the presence of the delta symbol within the discussion branch, we intentionally excluded the final comment awarding the delta to the user. We will make sure to clarify this point in the paper. However, since Figure 3 looks specifically at the start positions of ADUs, it might be that persuasive comments tend to have longer ADUs at the end. Although this hypothesis needs further verification and might be an interesting area for future research.
>
> __Response to Typos, Grammar, Style, and Presentation Improvements:__
>
> We appreciate the reviewer's recommendation to emphasize the significance of the new CMV dataset. In the revised paper, we will explicitly highlight the substantial time gap between existing datasets and ours, underscoring the importance of incorporating the latest content for a comprehensive analysis.
>
> __Question to the Reviewer:__
>
> The reviewer mentions the following:
> > The paper validates the edit and embeddings features with predictive performance, so this may be in the reader's mind as they read the paper.
>
> Could you kindly elaborate on what is meant by this? We would appreciate further clarification, as the identified strategies using the mentioned approaches can indeed be solely used for persuasiveness prediction.

---

### Meta-Review · Area_Chair_g4EB · 2023-09-18

**Recommendation:** 3

**Metareview:**

The work presents a dataset of Change My View (CMV) subreddit text annotated with argumentative discourse unit (ADU) typology by Morio et al. (2019). It argues that the ordering of the ADUs affect the persuasiveness and supports its claim with experimental results on determining whether a user has changed their view given a brand of comments.

Reviewers appreciated the study of ADU type arrangement patterns and strategies, leading to insights allowing tangible gains in persuasiveness prediction experiments. They also thought a new CMV dataset is good update to the existing one published in 2016.

A primary reviewer concern was the lack of experiment results using the latest models. While the authors provide additional results using LLaMA-2 (7B) in the rebuttal, I agree with the reviewers, and in fact the authors, that reframing the paper to focus on the analysis of argument arrangement will lead to a bigger impact in the community. Also, this work adopts the ADU typology from Morio et al. (2019). However, the typology is a minimally modified version of Park et al. (2015)’s typology. Consequently, the authors cite it, as well, for proper attribution.

---

### Decision · Program_Chairs · 2023-10-07

**Decision:**

Accept-Findings

**Comment:**

The work presents a dataset of Change My View (CMV) subreddit text annotated with argumentative discourse unit (ADU) typology by Morio et al. (2019). It argues that the ordering of the ADUs affect the persuasiveness and supports its claim with experimental results on determining whether a user has changed their view given a brand of comments.

Reviewers appreciated the study of ADU type arrangement patterns and strategies, leading to insights allowing tangible gains in persuasiveness prediction experiments. They also thought a new CMV dataset is good update to the existing one published in 2016.

A primary reviewer concern was the lack of experiment results using the latest models. While the authors provide additional results using LLaMA-2 (7B) in the rebuttal, I agree with the reviewers, and in fact the authors, that reframing the paper to focus on the analysis of argument arrangement will lead to a bigger impact in the community. Also, this work adopts the ADU typology from Morio et al. (2019). However, the typology is a minimally modified version of Park et al. (2015)’s typology. Consequently, the authors cite it, as well, for proper attribution.